# Ask not what AI can do, but what AI should do: Towards a framework of task delegability

Brian Lubars
University of Colorado Boulder
brian.lubars@colorado.edu

Chenhao Tan
University of Colorado Boulder
chenhao.tan@colorado.edu

## Abstract

While artificial intelligence (AI) holds promise for addressing societal challenges, issues of exactly which tasks to automate and to what extent to do so remain understudied. We approach this problem of task delegability from a human-centered perspective by developing a framework on human perception of task delegation to AI. We consider four high-level factors that can contribute to a delegation decision: motivation, difficulty, risk, and trust. To obtain an empirical understanding of human preferences in different tasks, we build a dataset of 100 tasks from academic papers, popular media portrayal of AI, and everyday life, and administer a survey based on our proposed framework. We find little preference for full AI control and a strong preference for machine-in-the-loop designs, in which humans play the leading role. Among the four factors, trust is the most correlated with human preferences of optimal human-machine delegation. This framework represents a first step towards characterizing human preferences of AI automation across tasks. We hope this work encourages future efforts towards understanding such individual attitudes; our goal is to inform the public and the AI research community rather than dictating any direction in technology development.

## 1 Introduction

Recent developments in machine learning have led to significant excitement about the promise of artificial intelligence. Ng [35] claims that "artificial intelligence is the new electricity." Artificial intelligence indeed approaches or even outperforms human-level intelligence in critical domains such as hiring, medical diagnosis, and judicial systems [7, 10, 12, 23, 29]. However, we also observe growing concerns about which problems are appropriate applications of AI. For instance, a recent study used deep learning to predict sexual orientation from images [45]. This study has caused controversy [32, 34]: Glaad and the Human Rights Campaign denounced the study as "junk science" that "threatens the safety and privacy of LGBTQ and non-LGBTQ people alike" [2]. In general, researchers also worry about the impact on jobs and the future of employment [14, 42, 44].

Such excitement and concern begs a fundamental question at the interface of artificial intelligence and human society: which tasks should be delegated to AI, and in what way? To answer this question, we need to at least consider two dimensions. The first dimension is capability. Machines may excel at some tasks, but struggle at others; this area has been widely explored since Fitts first tackled function allocation in 1951 [13, 36, 38]. The goal of AI research has also typically focused on pushing the boundaries of machine ability and exploring *what AI can do*.

The second dimension is human preferences, i.e., what role humans would like AI to play. The automation of some tasks is celebrated, while others should arguably *not* be automated for reasons beyond capability alone. For instance, automated civil surveillance may be disastrous from ethical, privacy, and legal standpoints. Motivation is another reason: no matter the quality of machine text generation, it is unlikely that an aspiring writer will derive the same satisfaction or value from

delegating their writing to a fully automated system. Despite the clear importance of understanding human preferences, the question of *what AI should do* remains understudied in AI research.

In this work, we present the first empirical study to understand how different factors relate to human preferences of *delegability*, i.e., to what extent AI should be involved. Our contributions are three-fold. First, building on prior literature on function allocation, mixed-initiative systems, and trust and reliance on machines [20, 27, 36, inter alia], we develop a framework of four factors — motivation, difficulty, risk, and trust — to explain task delegability. Second, we construct a dataset of diverse tasks ranging from those found in academic research to ones that people routinely perform in daily life. Third, we conduct a survey to solicit human evaluations of the four factors and delegation preferences, and validate the effectiveness of our framework. We find that our survey participants seldom prefer full automation, but value AI assistance. Among the four factors, trust is the most correlated with human preferences of delegability. However, the need for interpretability, a component in trust, does not show a strong correlation with delegability. Our study contributes towards a framework of task delegability and an evolving database of tasks and their associated human preferences.

## 2 Related Work

As machine learning grows further embedded in society, human preferences of AI automation gain relevance. We believe surveying, tracking, and understanding such preferences is important. However, apart from human-machine integration studies on specific tasks, the primary mode of thinking in AI research is towards automation. We summarize related work in three main areas.

**Task allocation and delegation.** Several studies have proposed theories of delegation in the context of general automation [6, 33, 36, 38]. Function allocation examines how to best divide tasks based on human and machine abilities [13, 36, 38]. Castelfranchi and Falcone [6] emphasize the role of risk, uncertainty, and trust in delegation, which we build on in developing our framework. Milewski and Lewis [33] suggest that people may not want to delegate to machines in tasks characterized by low trust or low confidence, where automation is unnecessary, or where automation does not add to utility. In the context of jobs and their susceptibility to automation, Frey and Osborne [14] find social, creative, and perception-manipulation requirements to be good prediction criteria for machine ability. Parasuraman et al.'s *Levels of Automation* is the closest to our work [36]. However, their work is primarily concerned with performance-based criteria (e.g., capability, reliability, cost), while our interest involves human preferences.

**Shared-control design paradigms.** Many tasks are amenable to a mix of human and machine control. Mixed-initiative systems and collaborative control have gained traction over function allocation, driven by the need for flexibility and adaptability, and the importance of a user's goals in optimizing value-added automation [4, 20, 21].

We broadly split work on shared-control systems into two categories. We find this split more flexible and practical for our application than the *Levels of Automation*. On one side, we have human-in-the-loop machine learning designs, wherein humans assist machines. The human role is to oversee, handle edge cases, augment training sets, and refine system outputs. Such designs enjoy prevalence in applications from vision and recognition to translation [5, 11, 18]. Alternatively, a machine-in-the-loop paradigm places the human in the primary position of action and control while the machine assists. Examples of this include a creative-writing assistance system that generates contextual suggestions [8, 41], and predicting situations in which people are likely to make judgmental errors in decision-making [1]. Even tasks which should not be automated *may* still benefit from machine assistance [16, 17, 24], especially if human performance is not the upper bound as Kleinberg et al. found in judge bail decisions [23].

**Trust and reliance on machines.** Finally, we consider the community's interest in trust. As automation grows in complexity, complete understanding becomes impossible; trust serves as a proxy for rational decisions in the face of uncertainty, and appropriate use of technology becomes critical [27]. As such, calibration of trust continues to be a popular avenue of research [15, 28]. Lee and See [27] identify three bases of trust in automation: performance, process, and purpose. *Performance* describes the automation's ability to reliably achieve the operator's goals. *Process* describes the inner workings of the automation; examples include dependability, integrity, and interpretability (in particular, interpretable ML has received significant interest [9, 22, 39, 40]). Finally, *purpose* refers to the intent behind the automation and its alignment with the user's goals.

| Factors | Components |
|---------|-----------|
| Motivation | Intrinsic motivation, goals, utility |
| Difficulty | Social skills, creativity, effort required, expertise required, human ability |
| Risk | Accountability, uncertainty, impact |
| Trust | Machine ability, interpretability, value alignment |

Table 1: An overview of the four factors in our AI task delegability framework.

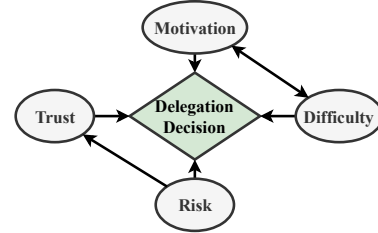

Figure 1: Factors behind task delegability.

## 3 A Framework for Task Delegability

To explain human preferences of task delegation to AI, we develop a framework with four factors: a person's **motivation** in undertaking the task, their perception of the task's **difficulty**, their perception of the **risk** associated with accomplishing the task, and finally their **trust** in the AI agent. We choose these factors because motivation, difficulty, and risk respectively cover why a person chooses to perform a task, the process of performing a task, and the outcome, while trust captures the interaction between the person and the AI. We now explain the four factors, situate them in literature, and present the statements used in the surveys to capture each component. Table 1 presents an overview.

**Motivation.** Motivation is an energizing factor that helps initiate, sustain, and regulate task-related actions by directing our attention towards goals or values [25, 30]. Affective (emotional) and cognitive processes are thought to be collectively responsible for driving action, so we consider *intrinsic motivation* and *goals* as two components in motivation [30]. We also distinguish between learning goals and performance goals, as indicated by Goal Setting Theory [31]. Finally, the expected *utility* of a task captures its value from a rational cost-benefit analysis perspective [20]. Note that a task may be of high intrinsic motivation yet low utility, e.g., reading a novel. Specifically, we use the following statements to measure these motivation components in our surveys:

1. **Intrinsic Motivation:** "I would feel motivated to perform this task, even without needing to; for example, it is fun, interesting, or meaningful to me."
2. **Goals:** "I am interested in learning how to master this task, not just in completing the task."
3. **Utility:** "I consider this task especially valuable or important; I would feel committed to completing this task because of the value it adds to my life or the lives of others."

**Difficulty.** Difficulty is a subjective measure reflecting the cost of performing a task. For delegation, we frame difficulty as the interplay between task requirements and the ability of a person to meet those requirements. Some tasks are difficult because they are time-consuming or laborious; others, because of the required training or expertise. To differentiate the two, we include *effort required* and *expertise required* as components in difficulty. The third component, *belief about abilities possessed*, can also be thought of as task-specific self-confidence (also called self-efficacy [3]) and has been empirically shown to predict allocation strategies between people and automation [26]. Additionally, we contextualize our difficulty measures with two specific skill requirements: the amount of *creativity* and *social skills* required. We choose these because they are considered more difficult for machines than for humans [14].

1. **Social skills:** "This task requires social skills to complete."
2. **Creativity:** "This task requires creativity to complete."
3. **Effort:** "This task requires a great deal of time or effort to complete."
4. **Expertise:** "It takes significant training or expertise to be qualified for this task."
5. **(Perceived) human ability:** "I am confident in [my own/a qualified person's] ability to complete this task." [1]

**Risk.** Real-world tasks involve uncertainty and risk in accomplishing the task, so a rational decision on delegation involves more than just cost and benefit. Delegation amounts to a bet: a choice considering the probabilities of accomplishing the goal against the risks and costs of each agent [6]. Perkins et al. [37] define risk practically as a "probability of harm or loss," finding that people rely on automation less as the probability of mortality increases. Responsibility or accountability may play a role if delegation is seen as a way to share blame [28, 43]. We thus decompose risk into three

components: personal ***accountability*** for the task outcome; the ***uncertainty***, or the probability of errors; and the scope of ***impact***, or cost or magnitude of those errors.

1. **Accountability:** "In the case of mistakes or failure on this task, someone needs to be held accountable."
2. **Uncertainty:** "A complex or unpredictable environment/situation is likely to cause this task to fail."
3. **Impact:** "Failure would result in a substantial negative impact on my life or the lives of others."

**Trust.** Trust captures how people deal with risk or uncertainty. We use the definition of trust as "the attitude that an agent will help achieve an individual's goals in a situation characterized by uncertainty and vulnerability" [27]. Trust is generally regarded as the most salient factor in reliance on automation [27, 28]. Here, we operationalize trust as a combination of perceived ***ability*** of the AI agent, agent ***interpretability*** (ability to explain itself), and perceived ***value alignment***. Each of these corresponds to a component of trust in Lee and See [27]: performance, process, and purpose.

1. **(Perceived) machine ability:** "I trust the AI agent's ability to reliably complete the task."
2. **Interpretability:** "Understanding the reasons behind the AI agent's actions is important for me to trust the AI agent on this task (e.g., explanations are necessary)." [2]
3. **Value alignment:** "I trust the AI agent's actions to protect my interests and align with my values for this task."

**Degree of delegation.** We develop this framework of motivation, difficulty, risk, and trust to explain human preferences of delegation. To measure human preferences, we split the degree of delegation into the following four categories (refer to Supplementary Material for the wordings in the survey):

1. **No AI assistance:** the person does the task completely on their own (henceforth, "human only").
2. **The human leads and the AI assists:** the person does the task mostly on their own, but the AI offers recommendations or help when appropriate (e.g., human gets stuck or AI sees possible mistakes) (henceforth, "machine in the loop").
3. **The AI leads and the human assists:** the AI performs the task, but asks the person for suggestions/confirmation when appropriate (henceforth, "human in the loop").
4. **Full AI automation**: decisions and actions are made automatically by the AI once the task is assigned; no human involvement (henceforth, "AI only").

Fig. 1 presents our expectation of how motivation, difficulty, risk, and trust relate to delegability. Motivation describes how invested someone is in the task, including how much effort they are willing to expend, while difficulty determines the amount of effort the task requires. Therefore, we expect difficulty and motivation to relate to each other: we hypothesize that people are more likely to delegate tasks which they find difficult (or have low confidence in their abilities), and less likely to delegate tasks which they are highly invested in. Risk reflects uncertainty and vulnerability in performing a task, the situational conditions necessary for trust to be salient [27]. We thus expect risk to moderate trust. Finally, we hypothesize that the correlation between components within each factor should greater than that across different factors, i.e., factors should show coherence in component measurements.

## 4   A Task Dataset and Survey Results

To evaluate our framework empirically, we build a database of diverse tasks covering settings ranging from academic research to daily life, and develop and administer a survey to gather perceptions of those tasks under our framework. We examine survey responses through both quantitative analyses and qualitative case studies.

### 4.1   A Dataset of Tasks

We choose 100 tasks drawn from academic conferences, popular discussions in the media, well-known occupations, and mundane tasks people encounter in their everyday lives. These tasks are generally relevant to current AI research and discussion, and present challenging delegability decisions with which to evaluate our framework. Example tasks can be found in §4.3. To additionally balance the variety of tasks chosen, we categorize them as art, creative, business, civic, entertainment, health, living, and social, and keep a minimum of 7 tasks of each (a task can belong to multiple

categories; refer to Supplementary Material for details). Ideally, the tasks would cover the entire automation "task space"; our task set is intended as a reasonable starting point.

Since some tasks, e.g., medical diagnosis, require expertise, and since motivation does not apply if the subject does not personally perform the task, we develop two survey versions.

- **Personal survey.** We include all the four factors in Table 1 and ask participants "If you were to do the given (above) task, what level of AI/machine assistance would you prefer?"
- **Expert survey.** We include only difficulty, risk, and trust, and ask participants "If you were to ask someone to complete the given (above) task, what level of AI/machine assistance would you prefer?"

Following a general explanation, our survey begins by asking subjects for basic demographic information. Subjects are then presented one randomly-selected task from our set. They evaluate the task under each component in our framework (see Table 1) according to a five-point Likert scale. Finally, subjects select the degree of delegation they would prefer for the task from the following four choices: Full Automation, AI leads and human assists, Human leads and AI assists, or No AI assistance. Note that subjects are not told which factor each question measures beyond the question text itself, and can choose the degree of delegation independently of our framework.

We administer this 5-minute survey on Mechanical Turk. To improve the quality of surveys, we require that participants have completed 200 HITs with at least a 99% acceptance rate and are from the United States. We additionally add two attention check questions to ensure participants read the survey carefully. Subjects are paid $0.80 upon completing the survey and passing the checks; otherwise the data is discarded. We record 1000 survey responses: 500 each for the personal and the expert versions, composed of 5 responses for each of the 100 tasks. Finally, we further discard responses that are identical in each component (e.g., marking "Agree" for all components), resulting in 495 and 497 responses for the personal and expert versions, respectively. This leaves 8 tasks with 4 responses rather than 5. We obtain a gender-balanced sample with 525 males, 463 females, and 4 identifying otherwise. The dataset is available at `http://delegability.github.io`.

## 4.2 Survey Results

In this section, we begin by examining the overall distribution of the delegability preferences, then investigate the relation between components in our framework and the delegability labels.

**Participants seldom choose "AI only" and prefer designs where humans play the leading role.** Participants labeled the delegability of tasks ranging from 1 ("Human only") to 4 ("AI only"). Fig. 2a presents the distribution. In both the *personal* and *expert* surveys, 4 ("AI only") is seldom chosen. Instead, both distributions peak at 2, indicating strong preferences towards machine-in-the-loop designs. This result becomes even more striking when averaging the five responses received per task, concentrating almost half the mass between 2 and 2.5 — again indicating a preference for machine-in-the-loop designs. In fact, after averaging responses, we find that none of the 100 tasks yield an overall preference for full automation ($\geq 3.5$). Taken together, these results imply that people prefer humans to keep control over the vast majority of tasks, yet are also open to AI assistance.

If we view our surveys as a labeling exercise, the agreement between individuals is low but is relatively higher in the *expert* survey than the *personal* survey: the Krippendorff's $\alpha$ is 0.063 in the *personal* survey and is 0.174 in the *expert* survey [19]. The lower agreement in the *personal* survey is consistent with heterogeneity between individuals. Two of the most contentious *personal* survey tasks were: "Cleaning your house" and "Breaking up with your romantic partner".

**Trust is most correlated with human preferences of automation.** Consistent with our hypothesis in Fig. 1, trust is generally positively correlated with delegability. Table 2 shows the component correlations with the delegability labels. After Bonferroni correction, 5 out of the 11 components are significantly correlated with the delegability label in the *expert* survey, while only 4 of the 14 components are in the *personal* survey. Trust takes the top two spots in both. Difficulty, the only other significantly correlated factor after trust, is generally negatively correlated with delegability. In particular, the creative and social skill requirements are the next highest correlations in both surveys, suggesting specific skills may form a stronger basis for delegation to AI than more subjective difficulty measures (e.g., self-confidence).

Next, we highlight three exceptions: 1) Contrary to our hypothesis, we did not find statistically significant correlations between the risk or motivation factors and delegability after Bonferroni cor-

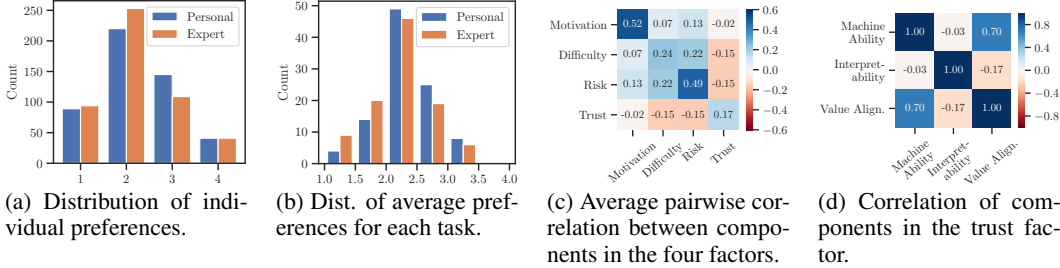

(a) Distribution of individual preferences.

(b) Dist. of average preferences for each task.

(c) Average pairwise correlation between components in the four factors.

(d) Correlation of components in the trust factor.

Figure 2: Fig. 2a and 2b show that full automation is rarely preferred in our surveys. Fig. 2c and 2d examine correlations between components. We observe low coherence in trust and difficulty. In particular, interpretability seems distinct from the other two trust components.

| Factor | Component | Personal | Expert |
|--------|-----------|----------|--------|
| Motivation | Utility | -0.126 (†) | N/A |
| Motivation | Intrinsic motivation | -0.104 (†) | N/A |
| Motivation | Goals | NS | N/A |
| Difficulty | Social skills | -0.303 (***) | -0.294 (***) |
| Difficulty | Creativity | -0.223 (***) | -0.290 (***) |
| Difficulty | Human ability | NS | -0.160 (**) |
| Difficulty | Effort required | NS | -0.124 (†) |
| Difficulty | Expertise required | NS | -0.120 (†) |
| Risk | Uncertainty | NS | -0.135 (†) |
| Risk | Accountability | NS | -0.123 (†) |
| Risk | Impact | NS | -0.106 (†) |
| Trust | Machine ability | 0.520 (***) | 0.593 (***) |
| Trust | Value alignment | 0.481 (***) | 0.522 (***) |
| Trust | Process | NS | NS |

Table 2: Pearson correlation of framework components with the delegabilty label for individual responses to the personal and expert surveys. $p$-values were computed by aggregating over individual Likert ratings separately for the *personal* and *expert* surveys, resulting in 25 tests in total. Significance after Bonferroni correction is indicated by *** for $p < 0.001$, ** for $p < 0.01$, * for $p < 0.05$, and NS for $p >= 0.05$. Results that were $p < 0.05$ prior to correction are indicated by †.

rection. 2) The *interpretability (process)* component of trust is not correlated with delegability. 3) Confidence in human ability (within the difficulty factor; lower confidence indicates higher difficulty) is not correlated with delegability in the *personal* survey, but is actually negatively correlated in the *expert* survey (the lower the confidence, the more delegable). This differs from the general trend of rating more difficult tasks as less delegable, suggesting that *people prefer experts to accept AI assistance on low-confidence tasks, but perhaps are less willing to do so themselves.*

**Factors are generally "coherent", but risk components have only weak correlation with trust components.** We next study the correlation between components in our framework. We focus on the *personal* survey here because it has all four factors, but results are consistent in the *expert* survey. Fig. 2c presents the average pairwise component correlations between the four factors: the correlation along the diagonal is generally higher than the off-diagonal ones. This finding confirms that factors are generally "coherent", affirming our categorization of the components within them.

Comparing the factor relations to our expectation in Fig. 1 (see the correlation graph in the Supplementary Material for detailed relations between components), we find that motivation and difficulty are indeed correlated, most notably between self-confidence and intrinsic motivation in the *personal* survey ($\rho = 0.41$, people enjoy doing what they are good at). However, contrary to our expectation, components in risk have only weak connections with trust, while difficulty is correlated with trust (through the social and creative skill requirements) and to risk in *personal* and *expert* surveys.

Coherence is lower in difficulty and trust than in risk and motivation. To investigate this, we zoom into the correlation matrix in Fig. 2d to show individual components of trust. We observe interpretability has little correlation with machine ability and negative correlation with value alignment, suggesting that the need for explanation is independent of whether the machine is perceived as capable, and perhaps higher when machine is perceived as benign. In comparison, interpretability is most strongly correlated with risk. In the *personal* survey, it is also connected to motivation through utility and learning goals. Thus risky and important tasks, which people want to learn, tend to require more interpretability; but this may not be tied to their willingness to delegate.

| Task Description | Social Skills (D) | Expertise Req (D) | Human Ability (D) | Account-ability (R) | Impact (R) | Machine Ability (T) | Interpret-ability (T) | Delegability |
|---|---|---|---|---|---|---|---|---|
| Medical diagnosis: flu | 3.4 | 4.2 | 4.6 | 4 | 4.2 | 3 | 4.2 | 2.4 |
| Medical treatment planning: flu | 3.6 | 3.6 | 4.4 | 4 | 3.4 | 3.6 | 3.8 | 2.4 |
| Explaining diagnosis & treatment options to a patient: flu | 3.4 | 3.6 | 4.2 | 3.8 | 3.2 | 3.8 | 3.2 | 2.2 |
| Medical diagnosis: depression | 4.4 | 4.6 | 4.6 | 4.2 | 3.8 | 2.8 | 3.4 | 2.2 |
| Medical treatment planning: depression | 3.8 | 3.4 | 4.4 | 4.2 | 4.4 | 3 | 4 | 2 |
| Explaining diagnosis & treatment options to a patient: depression | 4.4 | 4.4 | 4.2 | 4.4 | 4.4 | 2.2 | 4.4 | 1.6 |
| Medical diagnosis: cancer | 2.6 | 5 | 3.6 | 3.8 | 4.8 | 2.4 | 3.4 | 2 |
| Medical treatment planning: cancer | 3.6 | 4.6 | 4.8 | 4.4 | 4.8 | 2.4 | 3.8 | 1.6 |
| Explaining diagnosis & treatment options to a patient: cancer | 4.4 | 4.4 | 4.2 | 4.2 | 4.6 | 2.4 | 2.6 | <span style="color:red">1.4</span> |
| New employee hiring decisions | 4.4 | 3.6 | 3.8 | 3.8 | 3.8 | 2.4 | 4.4 | 2.2 |
| Judging a defendant's recidivism risk | 3.8 | 4 | 4.4 | 4.4 | 4.6 | 2.4 | 3.6 | 1.8 |

Table 3: A case study of tasks from the *expert* survey. See full names of tasks in the supplementary material. Note that we do not flip any component in these case study tables.

| Task Description | Social Skills (D) | Expertise Req (D) | Human Ability (D) | Account-ability (R) | Impact (R) | Intrinsic (M) | Machine Ability (T) | Delegability |
|---|---|---|---|---|---|---|---|---|
| Serving on jury duty | 4.6 | 2.4 | 4.6 | 4.6 | 4.2 | 3.8 | 1.2 | <span style="color:red">1.4</span> |
| New employee hiring decisions | 4 | 3.6 | 3.6 | 4.2 | 3 | 2.6 | 1.8 | 1.8 |
| Reading bedtime stories to your child | 4 | 2 | 4.4 | 2.6 | 1.8 | 4.8 | 3.2 | 1.8 |
| Scheduling an important business meeting | 3.6 | 2 | 4.4 | 3.2 | 3 | 2.6 | 3.6 | 3 |

Table 4: A case study of tasks from the *personal* survey. Refer to `https://delegability.github.io/table.html` for live demo.

## 4.3 Case Studies

To further illustrate the operation of our delegability framework, we present averaged responses to selected tasks from the *expert* survey in Table 3 and the *personal* survey in Table 4. For the expert case studies, we examine responses to medical diagnosis, recidivism risk, and hiring. Next, we observe the effects of motivation on some personal tasks. Finally, some tasks such as hiring are suitable for both experts and laypeople, allowing us to compare the personal and expert contexts. Though clearly important, we observe trust alone cannot explain differences in delegability preferences.

**Expert Survey Case Studies.** The medical domain is often considered a promising area for AI, but how open are patients to delegating different aspects of their healthcare to AI? We compare three medical tasks (diagnosis, treatment, and explanation), each with three different illness contexts (the flu, depression, and cancer). Intuitively, flu-related tasks are seen as the most delegable – even approaching a delegability of 3 (human-in-the-loop) – while cancer is the least. Correspondingly, the flu-related tasks are perceived as less difficult (lower social skill and expertise requirements, higher confidence in the human expert), less risky (lower impact and accountability), and with higher trust in machine ability. However, all except cancer explanation are nearest a delegability level of 2 (machine-in-the-loop): though human control is preferred, machine assistance is valued.

Fairness decision problems such as recidivism risk and employee hiring decisions are typically characterized as requiring high transparency and accountability. From the responses, we see that these tasks are both rated as difficult, risky, and with low degrees of trust in AI; in fact, they look similar to the medical tasks under our framework. Accordingly, we again observe both tasks result in an average preference for machine-in-loop designs.

**Personal Survey Case Studies.** To observe the role of motivation, we consider "Reading bedtime stories to your child", a machine-in-the-loop task, and "Scheduling an important business meeting with several co-workers", a human-in-the-loop task. The former is higher motivation, yet the latter is higher risk. The tasks are otherwise similar. Here, motivation appears to be the deciding factor: the former's higher motivation makes it less delegable despite the lower risk.

Finally, we compare some personal-context tasks which are similar to the above expert-context case studies. "Serving on jury duty: deciding if a defendant is innocent or guilty" is comparable to the high-risk low-trust medical and recidivism tasks, and is also similarly rated one of the least delegable tasks to AI. For the employee hiring task, respondents rated their self-confidence as near their confidence in an expert. Upon directly comparing the other components more closely, we observe higher accountability, lower trust, and slightly lower delegability levels in the personal evaluations.

# 5   Concluding Discussion

In this work, we present first steps towards understanding human preferences in human-AI task delegation decisions. We develop an intuitive framework of motivation, difficulty, risk, and trust, which we hope will serve as a starting point for reasoning about delegation preferences across tasks. We develop a survey to quantify human preferences, and validate the promise of such an empirical approach through correlation analysis and case studies. Our findings show a preference for machine-in-the-loop designs and a disinclination towards "AI Only".

In developing this framework, our intent is not to suppress technology development, but rather to provide an avenue for more effective human-centered automation. Human preferences regarding the extent of autonomy, and the reasons and motivations behind these preferences, are an understudied yet valuable source of information. There is a clear gap in methodologies for both understanding and contextualizing preferences across tasks; it is this gap that we wish to address.

**Implications.** First, our finding of trust as the most salient factor behind delegation preferences supports the community's widespread interest in trust and reliance. We find negative correlations between trust in machine abilities and the social and creative skill requirements. These are skills commonly considered difficult for machines, hinting that directly measuring task characteristics instead of human assessments of trust may also be an effective approach. We also note the *low* correlation between desired interpretability and delegability to AI, demonstrating the complex and intricate relation between interpretable machine learning and trust.

Moreover, our findings show that people do not prefer "AI only", instead opting for machine-in-the-loop designs. Interestingly, even for low-trust tasks such as cancer diagnosis or babysitting, people are still receptive to the idea of a machine-in-the-loop assistant. We should explore paradigms that let people maintain high-level control over tasks while leveraging machines as support, as in recent work on clinical decision systems [46].

**Limitations.** Our framework does not fully explain delegability preferences: the highest measured correlation is $0.59$, and due to limited data, we do not explore higher-order feature interactions. Additionally, human preferences are dynamic and survey results likely evolve over time. Nevertheless, mapping current perceptions enables tracking any future changes, providing a mechanism to understand how basic changes in factors like machine ability manifest through trust and reliance.

Our exploratory survey methodology also has several limitations. We abstracted delegability decisions and measured a limited number of factors, thus potentially overstating the importance of trust and overlooking others that occur in real situations. For example, we did not consider situational details like the trust in specific companies, or actual human or machine performance. Second, we designed the survey to avoid biasing or requiring respondents to understand our framework conceptually. Training may improve subject calibration and agreement. We also recommend consideration of individual baseline attitudes towards automation. Since each participant only filled out one survey, we were unable to determine if individuals were consistently biased towards AI, and what effect this might have had on our measurements. In addition, we chose only four delegability categories, ranging from "human only" to "AI only". This was a deliberate abstraction choice to handle the wide variety of tasks presented. However, since most responses fell into one of the two shared-control categories, future studies may benefit from more fine-grained choices on shared control.

Finally, our empirical survey is based on participants on Mechanical Turk. We use a strict filter and attention checks to guarantee quality, but this sample may not be representative of the general population in the US, or of other populations with different cultural expectations.

**Towards a Framework.** In addition to resolving the above limitations, we specifically suggest two characteristics for any framework addressing the task delegability question: 1) a characterization of human preferences towards automation and machine control; and 2) a characterization of the task space, enabling the generalization of task-specific findings to other domains.

In particular, generalization represents a significant challenge, especially when incorporating human factors. For instance, it is unclear how to generalize from physicians interacting with AI for cancer diagnosis to judges interacting with AI for recidivism prediction. While our approach maps tasks to delegability preferences through a common set of task-automation perception factors and enables directly comparing tasks with a quantifiable perception distance, its capability of generalization requires further verification. Ultimately, we believe an effective quantification of human preferences and task relations will prove invaluable for the community and the public as a whole.

## Footnotes

[1]We flip this component in coherence analysis (Fig. 2c) so that higher lack of *confidence in human ability* indicates higher difficulty.

[2]We flip this component in coherence analysis (Fig. 2c and 2d) so that higher lack of *need for interpretability* indicates higher trust.

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
