[Supplementary Material]

# Supplementary Material for "Ask Not What AI Can Do But What AI Should Do: Towards a Framework of Task Delegability"

Brian Lubars
University of Colorado Boulder
brian.lubars@colorado.edu

Chenhao Tan
University of Colorado Boulder
chenhao.tan@colorado.edu

## Component Correlations

Due to lack of space, we did not enumerate all correlations between components in the main paper, instead focusing mainly on the correlations between the factors and the task's delegability to AI. However, some interesting structure can also be seen in the connections between components themselves. Figure 1 shows the strongest component correlations ($|\rho| \geq 0.20$) for direct comparison of our expected framework connections to the observed connections.

(a) Component correlations in the *personal* survey.

(b) Component correlations in the *expert* survey.

Figure 1: Component correlations in the *personal* survey and *expert* survey. Both figures show only connections with correlation coefficient $|\rho| \geq 0.20$ to prevent overcrowding the graphic. The weight of the edge is proportional to $|\rho|$.

## Task Selection & Methodology

In selecting our dataset of 100 tasks, our aim is to create a diverse set that is relevant to current AI research and discussion. Ideally we would compile a large reference set that covers the entire automation "task space", but these 100 tasks are meant as a reasonable starting point. We source our task set from papers in AI conferences (96 tasks), from occupational descriptions (102 tasks),

from media coverage of AI (76 tasks), and from daily life (115 tasks). For example, "Analyzing and critiquing aesthetic qualities of photographs or other forms of art" is drawn from Chang et al. [1]. The occupational descriptions are adaptations of a subset of Frey and Osborne [2]'s dataset of 702 occupations, which were themselves originally adapted from O*NET. We select a subset which evenly spans the range of predicted occupational susceptibilty to automation [2].

To refine these 389 tasks down to 100 while promoting variety, we then group the tasks into 8 semantic categories: art, creative, business, civic, entertainment, health, living, and social. A task may belong to multiple categories. For instance, "Babysitting your child" is living and social. Our final set contains a minimum of 7 tasks per category. The 100 tasks, their sources, and their semantic categories are shown in Table 1.

| Task | Source | Categories |
| --- | --- | --- |
| Analyzing and critiquing aesthetic qualities of photographs or other forms of art | conference | art, creative |
| Choreographing dance moves for a person to perform | conference | art, creative |
| Drawing or painting something (making art) | life | art, creative |
| Picking a topic to write a short story about | life | art, creative |
| Reviewing a book or a movie | life | art, creative |
| Writing a blog post | life | art, creative |
| Writing a novel or a short story (creative writing) | occupation | art, creative |
| Analyzing and sorting legal documents for important information, e.g., to find legal precedents for arguing a case in court (similar to some of what a paralegal might do) | media | business |
| Analyzing financial market conditions and executing market orders for a large company (e.g. buy/sell stocks) | conference | business |
| Assembling automobiles in a factory | conference | business |
| Choosing and ordering food to eat for dinner | life | business |
| Coordinating and oversee construction of a building, e.g., consulting with engineers, surveyors, specialists, and construction workers – similar to some of what an architect might do. | occupation | business |
| Deciding which applicants receive a loan from a bank (loan assessment) | conference | business |
| Detecting and removing fake/deceptive online reviews (e.g., for hotels or products) | conference | business |
| Driving a truck delivering goods/cargo between cities | conference | business |
| Driving to work | conference | business |
| Establishing compensation/wage/salary level for an employee | occupation | business |
| Inferring damage for insurance purposes after a car accident | conference | business |
| Interviewing job applicants and rating candidates | media | business |
| Monitoring farm animals' (e.g., cows) behavior, predicting health issues, and alerting the farmer. | media | business |
| Moving & packing merchandise in a warehouse for shipping to customers | media | business |
| Picking jobs to apply to | conference | business |
| Planning menus and developing recipes at a restaurant | occupation | business |

| | | |
|---|---|---|
| Predicting the sale value of a real estate property | conference | business |
| Responding to emails at work | life | business |
| Scheduling an important business meeting with several co-workers | life | business |
| Serving food to customers at a restaurant | conference | business |
| Writing reports and publishing Olympic (or other sports) results, standings, and stats (sports news coverage) | media | business |
| Writing reports and publishing updates on House/Senate/gubernatorial races during election day (election news coverage) | media | business |
| Cutting, drying, and styling hair, similar to what a barber or hairstylist might do | occupation | business, creative |
| Designing new clothing to manufacture and sell (similar to what a fashion designer might do) | conference | business, creative |
| Finding products you might be interested in while you're shopping | conference | business, living |
| Teaching your child elementary school math (e.g., multiplication, fractions) | life | business, living, civic |
| Deciding which applicants to hire as new employees for an open position at work | conference | business, social |
| Taking photos of a planned event, such as a wedding or graduation, similar to what a professional photographer might do. | occupation | business, creative, social |
| Analyzing and controlling the flow of traffic in a city | conference | civic |
| Arguing your case when you're a defendant in a criminal court | life | civic |
| Deciding military actions such as whether to launch airstrikes | media | civic |
| Detecting/recognizing abnormal or suspicious activities of people in crowds in public places for the purposes of security and safety (similar to part of what a police officer might do) | conference | civic |
| Finding and rescuing survivors after earthquakes | conference | civic |
| Guiding and explaining exhibits in a museum (similar to what a museum tour guide might do) | conference | civic |
| Helping to locate a missing child by searching public spaces | media | civic |
| Identifying and flagging fake/deceptive news articles | conference | civic |
| Identifying and flagging online hate speech | media | civic |
| Identifying people who attended a political rally | media | civic |
| In court, determining a defendant's risk (e.g., in committing another crime or missing the court date), to help judges make decisions about bail, sentencing, or parole | media | civic |
| Responding to 911-police incident reports, similar to what a patrol officer might do | occupation | civic |
| Serving on jury duty: deciding if a defendant is innocent or guilty | life | civic |
| Setting tariffs on goods imported from China | media | civic |

| | | |
|---|---|---|
| Teaching a religion's doctrine and practices to followers, similar to some of the responsibilities of clergy/religious leaders | occupation | civic, social |
| Voting in federal elections | life | civic |
| Picking a movie to watch | conference | entertainment |
| Picking a movie to watch with a group of friends | life | entertainment |
| Picking songs to listen to | media | entertainment |
| Picking which advertisements to show to people on social media websites | media | entertainment |
| Picking which news stories to show to people on social media websites | media | entertainment |
| Playing a board game (e.g., monopoly, scrabble) | conference | entertainment |
| Playing a competitive game (e.g., dota2, starcraft, poker) | media | entertainment |
| Advising people on nutrition/their diet to help improve their health, similar to what a nutritionist might do | occupation | health |
| Conducting a risk prognosis assessment for deciding which patients to transfer to the ICU given limited resources (intensive care) | conference | health |
| Devising treatment plans for patients sick with the flu | conference | health |
| Devising treatment plans for patients with cancer | conference | health |
| Devising treatment plans for patients with depression | conference | health |
| Diagnosing whether a person has cancer | conference | health |
| Diagnosing whether a person has depression | conference | health |
| Diagnosing whether a person has the flu | conference | health |
| Explaining the diagnosis and treatment options for the flu to a patient | occupation | health |
| Explaining the diagnosis and treatment options of cancer to a patient | occupation | health |
| Explaining the diagnosis and treatment options of depression to a patient | occupation | health |
| Helping stroke patients with physical rehabilitation, by guiding or assisting with exercise motions when needed (similar to what a physical therapist might do as part of their job) | conference | health |
| Monitoring your health and alerting when you should go to the doctor | media | health |
| Providing and coordinating patient care in a health facility, similar to a small part of what a Registered Nurse might do. | occupation | health |
| Assisting an elderly person with showering or bathing | conference | living |
| Brushing your teeth | life | living |
| Buying groceries | life | living |
| Cleaning up toxic waste, e.g., after a chemical spill | conference | living |
| Cleaning your house | life | living |
| Cooking dinner | life | living |
| Deciding on an outfit for you to wear | conference | living |

| | | |
|---|---|---|
| Describing images or scenes for visually impaired people | conference | living |
| Editing an internet forum comment before you post it (e.g., for maximum popularity) | life | living |
| Filling out and submitting your federal tax return paperwork | life | living |
| Managing your personal finances/investments (similar to what a financial advisor might do) | life | living |
| Monitoring a person's driving and intervening when they're distracted/in danger of making a mistake (e.g., emergency braking) | conference | living |
| Tracking important moments and information and creating memory aids for elderly people | conference | living |
| Translating an article you'd like to read from a foreign language to English | conference | living |
| Asking a person out on a date | life | living, social |
| Assisting your elderly parent | life | living, social |
| Babysitting your child | life | living, social |
| Breaking up with your romantic partner | life | living, social |
| Finding people who might like to meet for a date | life | living, social |
| Helping elderly individuals to increase their mobility by guiding them through crowded public spaces (e.g., walking to the grocery store) | conference | living, social |
| Identifying the social relationship between two people (e.g., are they friends, a couple, strangers, siblings) | conference | living, social |
| Picking out and buying a birthday present for an acquaintance | living | living, social |
| Predicting the sexual orientation of a person | media | living, social |
| Reading bedtime stories to your child | media | living, social |
| Telling a joke | life | living, social |
| Thinking of conversation topics while hanging out with friends | life | living, social |
| Walking your dog | life | living, social |
| Writing a birthday card to your mother | life | living, social |

Table 1: The set of 100 tasks presented in our surveys.

## Survey Administration

We advertise the survey on Amazon Mechanical Turk as a HIT (Human Intelligence Task) to workers who meet our quality screening guidelines. Specifically, participants must have completed 200 HITs with at least a 99% acceptance rate and must be from the United States. Before accepting the HIT, participants are presented with the IRB-approved informed consent information, including the compensation amount ($0.80) and a brief description of the survey and its purpose. Upon providing informed consent and accepting the HIT, participants are presented with our survey. Participants are only permitted to accept our HIT one time.

Upon accepting the HIT, participants are first shown the survey instructions, then the demographic questions. Next, participants are presented with one randomly-selected task. Participants evaluate the task under each component in our framework according to a five-point Likert scale. Two attention questions are mixed in to this section. Finally, participants choose the degree of delegation they would prefer for the task. The questions are presented in a fixed order (not randomized). Subjects are

paid $0.80 upon completing the survey and passing the checks attention check questions; otherwise the data is discarded. Note that participants are not told which factor each question measures beyond the question text itself, and can choose the degree of delegation independently of our framework. The HIT takes approximately 5 minutes to complete. The full survey text is given in the next section.

## Survey Questions & Demographics

### Survey Instructions

Note: You may only complete ONE HIT. Please do not queue hits or you will slow down study completion and delay payment.

We are conducting an academic survey about peoples attitudes towards delegating different kinds of tasks to an AI (artificial intelligence) versus to a person. You will:

Provide basic demographic information Offer your opinion on properties of a task (e.g., mowing a lawn) in the form of agree/disagree statements Choose the best way to divide control of the task between an AI and a person.

We expect the survey to take approximately 5-10 minutes, and you will be compensated $0.80 upon submission (Expect approval within 1-2 days; Please note, your submission may not be approved if the attention questions are not answered correctly).

### Demographic Questions

The following questions will help us to understand the study population and representativeness.

#### 1. What best describes your gender?

- Male
- Female
- Prefer to self-describe:
- Prefer not to say

#### 2. What is your age?

- 18-25
- 26-35
- 36-45
- 46-55
- 56-65
- 66-75
- 76 or older
- Prefer not to say

#### 3. How would you rate your level of computer proficiency?

- Far above average
- Slightly above average
- Average
- Slightly below average
- Far below average
- Prefer not to say

#### 4. What is the highest degree or level of school you have completed? (If currently enrolled, highest degree received)

- Some high school, no diploma, and below
- High school graduate, diploma or equivalent (for example: GED)
- Some college credit, no degree
- Trade/technical/vocational training
- Bachelor's degree, and above
- Prefer not to say

**Personal Survey Questions**

The following questions are the primary focus of this study.

**Important:** We will display a task. When answering the following questions, please carefully consider the task and your beliefs about an AI (artificial intelligence) agent performing the task, versus you personally performing the task.

(What is an AI agent? You can think of it as a computer, machine, robot, or some other form of automation.)

**Here is the task:** <Task>

**5. For each of the following statements, please indicate the extent to which you agree or disagree for the above task:**

1. This task requires social skills to complete.
2. This task requires creativity to complete.
3. This task requires a great deal of time or effort to complete.
4. Attention check, please choose 'Strongly Agree' for this one.
5. It takes significant training or expertise to be qualified for this task.
6. I am confident in my own abilities to complete this task.
7. In the case of mistakes or failure on this task, someone needs to be held accountable.
8. A complex or unpredictable environment/situation is likely to cause this task to fail.
9. Failure would result in a substantial negative impact on my life or the lives of others.
10. I would feel motivated to perform this task, even without needing to; for example, it is fun, interesting, or meaningful to me.
11. I am interested in learning how to master this task, not just in completing the task.
12. I consider this task especially valuable or important; I would feel committed to completing this task because of the value it adds to my life or the lives of others.
13. Attention check, please choose 'Strongly Disagree' for this one.
14. I trust the AI agent's ability to reliably complete the task.
15. Understanding the reasons behind the AI agent's actions is important for me to trust the AI agent on this task (e.g., explanations are necessary).
16. I trust the AI agent's actions to protect my interests and align with my values for this task.

**6. If you were to do the given (above) task, what level of AI/machine assistance would you prefer?**

1. **Full AI automation:** decisions and actions are made automatically by the AI once the task is assigned; you do nothing.
2. **The AI leads and the human assists:** the AI performs the task, but asks you for suggestions/confirmation when appropriate.
3. **The human leads and the AI assists:** you do the task mostly on your own, but the AI offers recommendations or help when appropriate (e.g., you get stuck or AI sees possible mistakes).
4. **No AI assistance:** you do the task completely on your own.

**Expert Survey Questions**

The following questions are the primary focus of this study.

**Important:** We will display a task. When answering the following questions, please carefully consider the task and your beliefs about an AI (artificial intelligence) agent performing the task, versus a qualified human person performing the task.

(What is an AI agent? You can think of it as a computer, machine, robot, or some other form of automation.)

**Here is the task:** <Task>

5. **For each of the following statements, please indicate the extent to which you agree or disagree for the above task:**

1. This task requires social skills to complete.
2. This task requires creativity to complete.
3. This task requires a great deal of time or effort to complete.
4. Attention check, please choose 'Strongly Agree' for this one.
5. It takes significant training or expertise to be qualified for this task.
6. I am confident in a qualified person's ability to complete this task.
7. In the case of mistakes or failure on this task, someone needs to be held accountable.
8. A complex or unpredictable environment/situation is likely to cause this task to fail.
9. Failure would result in a substantial negative impact on my life or the lives of others.
10. Attention check, please choose 'Strongly Disagree' for this one.
11. I trust the AI agent's ability to reliably complete the task.
12. Understanding the reasons behind the AI agent's actions is important for me to trust the AI agent on this task (e.g., explanations are necessary).
13. I trust the AI agent's actions to protect my interests and align with my values for this task.

6. **If you were to ask someone to complete the given (above) task, what level of AI/machine assistance would you prefer?**

1. **Full AI automation:** decisions and actions are made automatically by the AI once the task is assigned; no human involvement.
2. **The AI leads and the human assists:** the AI performs the task, but asks the person for suggestions/confirmation when appropriate.
3. **The human leads and the AI assists:** the person does the task mostly on their own, but the AI offers recommendations or help when appropriate (e.g., human gets stuck or AI sees possible mistakes).
4. **No AI assistance:** the person does the task completely on their own.

**Demographics.** We have two surveys (delegating to experts vs AI, or the subject personally vs AI), 100 tasks per survey, and 4 or 5 responses per task. Of the 992 subjects (495 in the *personal* survey and 497 in the *expert* survey), 525 identified as male, 463 as female, 2 as non-binary, and 2 preferred not to indicate. 136 were aged 18-25, 421 aged 26-35, 223 aged 36-45, 118 aged 46-55, 78 aged 56-65, 13 aged 66-75, 1 aged 76+, and 2 preferred not to indicate.