[Reviews · NeurIPS 2019]

Reviewer 1



* Summary This work presents a framework for task delegability and empirically investigates through a survey people's preferences for delegating different tasks to AI. This work is both theoretical and empirical. The theoretical part concerns the framework, whereas the empirical part is an analysis of the survey responses, mostly through correlation analysis. * Originality To the best of my knowledge the presented framework for task delegability is novel (the authors state that the framework builds on prior ideas in literature). The authors state that this is the first empirical study on task delegability. The authors provide a quite well-structured related work section which appears to cover a diverse list of related topics. The authors also highlight some work related to the present work and point out differences. * Quality and Significance The technical quality of the theoretical part of the paper is quite good. I think that the delegability framework is presented clearly and motivations are also provided for the various components in the framework. i find that the the empirical part of the paper is technically OK. The methodology in the empirical evaluation is basically based on correlation analysis. Some things could be clarified in the empirical part. - Line 224: be more specific and say that the agreement is between individuals - Table 2: mention how the correlations are calculated, i.e., what you have aggregated over - Table 2: please do not omit components that are always non-significant, this is also interesting information. - Table 2: Are p-values corrected for multiple comparisons? In their concluding discussion, the authors point out some limitations regarding their work, and this is good. However, it would also be important for the authors to include some critical discussion concerning the effect size of the observed correlations (the correlations are quite low, even for the significant findings). In overall, I find that the theoretical part of the paper (introduction of the framework) is quite OK. The empirical part seems a bit simplistic given the nice dataset that the authors have collected and some kind of more detailed analysis could be warranted here. Some of the conclusions seem quite "self-evident", but this is Ok since the results are based on the empirical findings. The case-studies did not really convey that much extra information, I think. One must also ask how relevant and important these findings are, i.e., what is the potential significance. I am not sure that these results have that high practical relevance and hence the significance of this study might not be that high. I am also not sure that this paper is a good fit for NeurIPS, since the this study is more "sociologic" in nature than computer science (although the topic of study, naturally, is related to computer science). * Clarity The structure and language of this paper is OK and the ideas are clearly communicated. * Update after Author Feedback I have now read the author feedback. The authors' responses addressed some of the questions I had concerning the results. Also, I now think that this paper does fit the conference. I have hence increased my rating of this paper.

Reviewer 2



The problem this paper studies, determining whether or not to deploy AI tools, is a vital and under-studied problem, and it's a pleasure to see here. I agree that understanding current attitudes is an important part of the equation, although I'm not particularly familiar with the task delegability literature in general. It's hard to tell how these survey responses are influenced by participants' assumptions about both the type/quality of AI assumed and who built the AI. I worry what this survey is actually measuring is people's assumptions about who is doing the building, and not what they are building. If they are assuming this is Google building the AI and imbuing it with its values rather than, say, the doctors doing the diagnosis, I imagine this should affect their ratings of both risk and trust. It's harder to hold Google accountable than your doctor, for example. This is particularly pronounced in value alignment, where it depends on the participants' assumptions about how or even if the AI was imbued with values. Do the respondents assume that, contrary to reality, AI is value neutral? In addition, while this work concludes that trust is the most important factor, I am concerned that this was in part driven by phrasing like "I trust the AI agent's ability to reliably complete the task," which could easily be mistaken for just whether the task should be delegated. 'Ability' could easily be interpreted to include the difficulty of the task. This would explain the high correlation between the decisions and the trust component. Finally, one component that appears to be missing is how much a time or effort the *participant* spends on the task. If they're not annoyed by having to do the task themselves, then there's less incentive for them to want to automate it. Other concerns: -192: Why not just ask the actual experts, instead of asking people for their guesses of what experts are capable of? -Table 2: How are the p-values being computed? This is an example of multiple-hypothesis testing, and claims of significance should of course reflect that. -265: typo Edit: I acknowledge the response to my review and while I still believe that this work would be made stronger if it had attempted to understand the kinds of assumptions that people make about AI, e.g. how they're built, it's fine if that's future work.

Reviewer 3



The paper presents an analysis of human preferances and understandings of AI performing tasks at four levels of delegation (From fully autonomous to not present at all). The idea is to develop a framework that can be used to gain better understandings of how humans perceive AI. This kind of method could be used to gather insights as AI systems become more capable into how humans perceive them and are willing to use them. The paper is well written for the most part and clear. I think the results are significant, and the breakdown in terms of the components is very interesting. The case studies further accentuate the points presented. This kind of paper will be of interest to artificial intelligence policy makers. I have a few suggestions and questions, but I think most of these are easily addressable. - in section 3 its not clear (although it is directly stated) that the items shown in numbered lists are the actual questions asked - maybe put them in "quotes" or italics to highlight this better. - the last paragraph of section 3 threw me off with the "four factors" which I thought were the 4 in the list right above it. This section could be reorganized a bit to handle this and the previous comment - the choice of tasks needs more description. "100 tasks drawn from academic conferences" is not clear, and a better description of the criteria used to select the tasks is needed. If there were not inclusion criteria used then how was this done? - were the questions about the properties of tasks randomized across participants? If not, why not? - was the delegation question always asked after the task questions? Would anything change if it was asked first? - give a few small examples of tasks in the main text at the start of 4.1 to help situate the reader. - in 4.1 - ""We include all the four factors" - again not clear what this is referring to - be specific. - in 4.1 it wasn't clear on the first read through that each person just evaluated a single task - actually I'm still not sure of this but think so given that each survey took 5 minutes. IN general the procedure could be clarified and it could be made explicit which questions were asked (e.g. number the questions in section 3 as M.1,M.2 for Motivation, D.1,D.2 for Difficulty and then refer to these explicitly in 4.1 - in 4.2 the "degree of AI assistance" -- make it clear this is the delegatability - or use one term only consistently throughout - in 4.2 the notion of "coherent" is not defined - in 4.2 explain the high correlation between value alignment and machine ability - I found tables 3 adn 4 rather hard to look at - it might help to color code each table cell to show larger/smaller - the numbers with decimals are hard to easily compare at a glance. i have read the author's response and it clarifies the issues I raised.

[Author Response · NeurIPS 2019]

We thank all the reviewers for their reviews! We will address the excellent writing/presentation-related suggestions in revision. Here we focus on clarifying questions about the framework, surveys, and results.

**Re R1: Relevance and significance of results.** We believe that the paper is appropriate for NeurIPS because it speaks to the applications of AI. Notably, we find that most people prefer machine-in-the-loop designs, and trust (except interpretability) is highly correlated with human preferences of delegability. We believe that working towards an improved and fine-grained understanding of public perception of AI on different tasks (both now and in deltas as the technology grows), will be valuable for researchers, industry leaders, and policy makers alike.

**Re R1 & R2: Pearson r & p-value calculations in Table 2.** Correlations in Table 2 are calculated by aggregating over individual likert ratings of the given component and the delegability level, for each individual response. This is done separately for the *personal* and *expert* surveys, resulting in a total of 28 tests. We did not apply Bonferroni correction because the dataset is relatively small and we chose to present the original results as an exploratory demonstration. After applying Bonferroni correction (14 tests each in the *personal* and *expert* surveys = 28 tests), most of the * and **s become non-significant. We will clarify this correction and follow R1's suggestion to include non-significant components in the appendix. Note that our empirical analysis in the main paper is limited by space constraints.

**Re R1: Effect size.** Our most significant findings have substantial effect size: 73.3% subjects prefer mixed-control, 18.4% prefer human only, and only 8.3% prefer complete automation; trust is the factor most correlated with delegability ($\rho \sim 0.5$). While we do discuss how the other effects are weaker than we expected in section 4.2, we could add some concluding discussion as well. The lack of clear effects also speaks to the challenge and lack of existing literature around this problem: we hope future studies can build on these results to help explain delegability more completely.

**Re R2: Subjects' assumptions about AI/builders.** Peoples' evaluations of trust undoubtedly depend on unmeasured assumptions, like who built the AI and why. Our goal is to advance our understanding of people's preferences towards delegation to AI, implicit assumptions included, because these same hidden assumptions are also present in current discussions about AI. While enumerating assumptions such as accuracy, financial costs, and builders of AI may lead to interesting results (given that we observe a significant correlation of value alignment), we leave that to future work.

**Re R2: Conflation of the machine ability question with difficulty/delegability.** First, we believe that human evaluation of trust in machine ability *does* include an implicit evaluation of difficulty for machines by definition, as it estimates the machine's ability to complete the task. Second, trust, difficulty, and delegability are all nebulous concepts; our work attempts to break down specific aspects of them. Third, machine ability does not fully explain delegability preferences ($\sim 0.5$ in Table 2; lower than we would expect if people are conflating the two questions), and is at most weakly correlated with difficulty questions (see Figure 1 in Supplementary). For example, for "Reading bedtime stories to your child" in Table 3, we see reasonable trust in machine abilities (3.2/5), yet low delegability (1.8/4).

**Re R2: Missing time/effort component.** This was captured by the Effort component within Difficulty: "This task requires a great deal of time or effort to complete."

**Re R2: Ask actual experts.** Our primary interest is in understanding public preferences to delegate, so we administrated our surveys on Mechanical Turk, which does not typically involve experts. Also, given our diverse tasks, this would require experts for each task. That said, comparing expert-public differences is a great future direction.

**Re R2: Knowledge of AI.** We did not survey AI knowledge explicitly, but asked about computer knowledge. Subjects mostly rate themselves average/above, and there was no significant difference between them in delegability preferences.

**Re R3: Choice of tasks.** We source our task set from papers in AI conferences, daily life, occupations, and media coverage of AI. For example, "Analyzing and critiquing aesthetic qualities of photographs or other forms of art" comes from "Aesthetic Critiques Generation for Photos", ICCV 2017. These 100 tasks are meant as a first study; ideally, we build a large reference set that covers the entire "task space". Here the chosen set constitutes a reasonable starting point.

**Re R3: Randomization of survey questions measuring components, order of delegability question, and survey procedure.** The questions were not randomized, and the delegation question was always asked last. Each subject evaluated only one task. This ordering likely forced the subject to consider aspects of task delegability which they otherwise might not have, which could influence the decision, hopefully adding some depth. Because of the limited number of subjects (5 per task), question randomization may have introduced too much variability.

**Re R3: Notion of coherence.** By factor coherence, we mean the extent to which components within the factor may share information, as indicated by average pairwise correlations. High coherence justifies talking about the factors at a higher level, while low coherence indicates that more focus on individual components is warranted.

**Re R3: High correlation between Value Alignment (VA) and Machine Ability (MA) in Trust.** A nice future direction and we will add a discussion! Could be: evaluations of MA implicitly include VA; or VA is not considered an issue for tasks which currently have high MA; or an unknown component (e.g., builders of the AI from R2).

[Meta-Review · NeurIPS 2019]

Reviewers agree that this paper asks important question on what tasks humans think AI should and should not do. Reviewers found author's response helpful and it clarified a couple of points that reviewers have raised. Please edit the final version with reviewers' comments.